# Characterizing the referral care continuum among complex obstetric patients in the Blantyre District of Malawi: A mixed methods study

Ashley Mitchell [1,2]*, Luseshelo Simwinga[2,3], Alden Blair[1,2], Miranda Rouse[1,2], Kimberly Baltzell[1,2,4], Richard Malirakwenda[5], Joyce Jere[3], Oveka Mwanza[3]

1 Institute for Global Health Sciences, University of California, San Francisco, San Francisco, California, United States of America, 2 Global Action in Nursing (GAIN), University of California, San Francisco, San Francisco, California, United States of America, 3 GAIA Global Health, Blantyre, Malawi, 4 Department of Family Health Care Nursing, University of California, San Francisco, San Francisco, California, United States of America, 5 Seed Global Health, Blantyre, Malawi

* ashley.mitchell@ucsf.edu

## Abstract

Despite high rates of facility delivery and skilled attendance at birth in Malawi, maternal mortality remains high underscoring the need to improve quality of care. With most deaths occurring at secondary and tertiary levels of care, our midwifery-led team aimed to understand the characteristics of pre-referral care driving poor maternal outcomes. We leveraged a convergent, parallel mixed methods approach to characterize obstetric care and outcomes across the referral care continuum and explore facilitators and barriers to the referral process between March 2019 and March 2020. Complex obstetric patient charts from seven Blantyre District primary health centers referred to the only local tertiary hospital were extracted and analyzed for associations between individual- and facility-level characteristics and referral care and outcomes. Transcripts from three focus group discussions and 18 in-depth interviews with clinical providers and referred mothers were analyzed using qualitative inference. Among 398 birthing women, 54% were between 18 and 24 years-old and 32% were referred from a facility more than 10km from the Hospital. Compared to survivors, mothers who died (n = 10) were significantly more likely to have been referred from a facility >10 km away, to have arrived in critical condition, and 100% experienced complications during their stay ranging from postpartum hemorrhage to cardiac arrest. Three primary themes emerged as barriers to quality pre-referral care in the district: systemic and structural challenges, inconsistent inter- and intra-facility communication, and community and provider influences on maternal expectations and beliefs. Triangulation of these findings suggests that strengthening referral infrastructure, bolstering communication and documentation, and reducing total referral time are key to improving care quality and outcomes for complex obstetric cases.

**Data availability statement:** The data supporting this manuscript was made available to the authors by the National Health Sciences Research Committee (NHSRC), Queen Elizabeth Central Hospital (QECH) in Malawi, and the Malawi Ministry of Health. The data are not publicly available but may be requested from the NHSRC. Requests for access can be directed to the Chairperson on duty at mohdoccentre@gmail.com.

**Funding:** This work was supported by the Wyss Medical Foundation (Funding provided to GAIN UCSF). The funders had no role in study design, data collection and analysis, decision to publish, or preparation of the manuscript.

**Competing interests:** The authors have declared that no competing interests exist.

Additionally, providers across the care continuum need additional training and support to ensure timely interventions and comprehensive, continuous referral care.

## Introduction

Malawi has made significant strides to improve maternal health care and recently achieved high rates (>90%) of both facility delivery and skilled attendance at birth [1]. Skilled attendants are primarily nurse-midwives who provide 72% of antenatal care and 68% of care from the onset of labor through the delivery of the placenta ("intrapartum care") in the country [1]. Still, opportunities to improve maternal and neonatal outcomes remain as evidenced by a maternal mortality ratio (MMR) of 349 maternal deaths per 100,000 live births and a neonatal mortality rate (NMR) of 19 per 1,000 pregnancies as of 2019 [2]. While these figures are lower than the regional averages for MMR (545 per 100,000) and NMR (22 per 1,000), they are far from the global targets set by the United Nations 2030 agenda to achieve an MMR of less than 70 per 100,000 live births and an NMR of 12 per 1,000 live births [3–5].

High mortality rates, despite high rates of skilled attendance at birth, underscore that improvements in maternal and neonatal health cannot solely rely on getting women to facilities but also must address health care quality and systemic factors which influence obstetric care and outcomes. Research at secondary and tertiary hospitals in Lilongwe District, Malawi identified that, among cases resulting in maternal death, pre-referral challenges including missed and misdiagnosis, incomplete clinical documentation, and delayed transfers contributed to an increased risk of otherwise preventable mortality [6]. Additional research in southern and central Malawi found that improved monitoring and communication surrounding referrals strengthened quality of care and patient outcomes [7,8]. These studies recommended simple structural reinforcement, including the repair of nonfunctioning health facility radios, to prevent morbidity and mortality [7,8]. Exploration of barriers and bottlenecks across other health facilities and geographical areas could lead to similarly targeted solutions. Informed approaches would be likely to support quality care from the time a decision to refer is made until the patient receives needed care at the receiving facility—across the "referral care continuum".

In fact, a well-functioning referral system is continually highlighted by the global safe motherhood community as a critical component to prevent maternal morbidity and mortality [9–14]. Nursing research in high-income contexts has informed robust practices and policies for effective referrals [15–17]. A recent systematic review of referral systems research in low-income contexts suggested that relatively few studies have been conducted in these settings [18]. Still, existing literature describes complex and multi-faceted challenges involving personnel and economic resources that negatively affect referral systems [18]. Recent research from Malawi aligns with these findings and recommends health system reform to prevent the financial and human costs that result from delays surrounding maternal referrals [12,19,20]. However, there is a dearth of evidence around the design and implementation of

interventions to improve referrals in low-income settings [21]. Additionally, there is a lack of national efforts to address referral care issues in Malawi.

According to the Malawian National Committee for Confidential Enquiry into Maternal Deaths (NCCMD), over a quarter (27%) of maternal deaths in 2015 occurred in the southwest region of Malawi where there is only one tertiary hospital ("the Hospital") [22]. Of the 52 maternal deaths that occurred at the Hospital in 2019, 58% were referrals from the Blantyre District Health Office (DHO) health centers. Maternal risks are speculated to be increased by the high volume of referrals to a single hospital resulting in congestion of the maternity wards, high burnout rates among providers, and ultimately worsened care quality and outcomes [23,24]. Overburdened facilities are further limited by an insufficient number of skilled providers, exacerbating delays in the provision of emergency obstetric care [25].

To better understand the state of referrals and maternal deaths in Blantyre District, our binational team conducted this study. Our approach was also shaped by the desire to develop a midwife-led bundle of communication tools for use in peripheral facilities to accompany referred patients to the Hospital. In addition to informing local efforts to strengthen maternal referrals, this study aimed to contextualize the downstream health outcomes of existing care gaps and to inform solutions in settings with limited resources.

## Materials and methods

This study utilized a convergent, parallel mixed methods approach to characterize obstetric care and outcomes across the referral care continuum and explore facilitators and barriers of the referral process. The study serves as the basis for the development of an intervention package to address poor-quality pre-referral care, communication, and documentation for facilities referring to the Hospital. Quantitatively, the study linked patient charts at the Hospital with their referring facility to understand pre-referral care. Qualitatively the study interviewed clinicians, nurse-midwives, and patients to assess their experiences with the referral process. These were analyzed in tandem to identify areas in the referral process for potential intervention.

### Study setting

Since 2018, the Global Action in Nursing (GAIN) project at the University of California, San Francisco (UCSF) has partnered with local government and non-profit organizations in Malawi to elevate quality nursing and midwifery care through trainings and longitudinal bedside mentorship [26]. GAIN's work began in Neno District in 2017 and expanded to Blantyre District in 2019 where the Malawian government selected seven primary health care facilities to partner with GAIN (Facility A, Facility B, Facility C, Facility D, Facility E, Facility F, and Facility G) (Fig 1). These sites were chosen based on their high caseload of maternal and neonatal patients. All the facilities refer complex and emergency obstetric patients to the Hospital.

### Methodological approach

To quantitatively assess pre-referral care practices, patient charts of any woman referred from one of seven referring facilities to the Hospital between March 2019 – March 2020 were extracted and reviewed. This baseline cutoff was chosen in part to avoid confounding in the care process involving national shutdowns during the COVID pandemic. All referred patients from the facilities during this period were considered eligible for inclusion in the study. A subset of women for whom referral interval times could be calculated were included in a time-to-provider analysis (Fig 2). We use the term "referral interval" throughout to refer to the time between the identification of a complication prompting a referral and the time at which that patient was seen by a provider at the Hospital—this temporal period falls within the broader referral continuum concept. A sub-analysis involving maternal charts which included time/date of both events was conducted. Based on the average referral rate from seven facilities to the Hospital, accounting for potential confounding, and ensuring

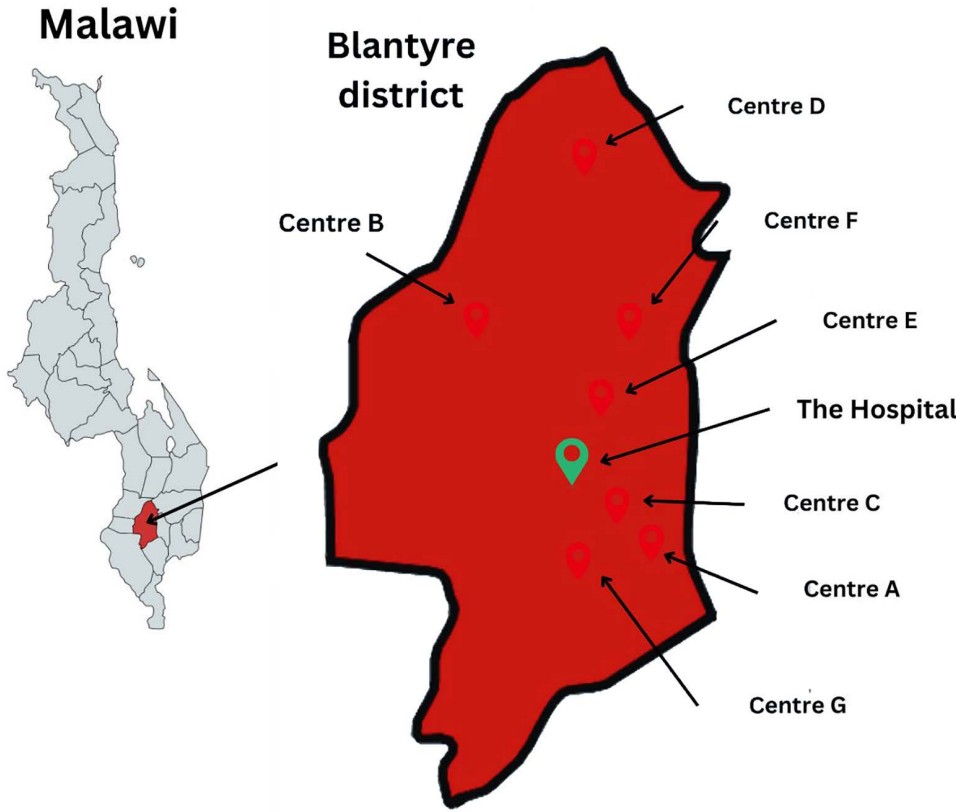

**Fig 1. Map of the Blantyre District health facilities included in the study in relation to the Hospital.** (*Adapted and republished from mapchart.net under a CC BY license, with permission from owner/founder Minas Giannekas, original copyright 2024*).

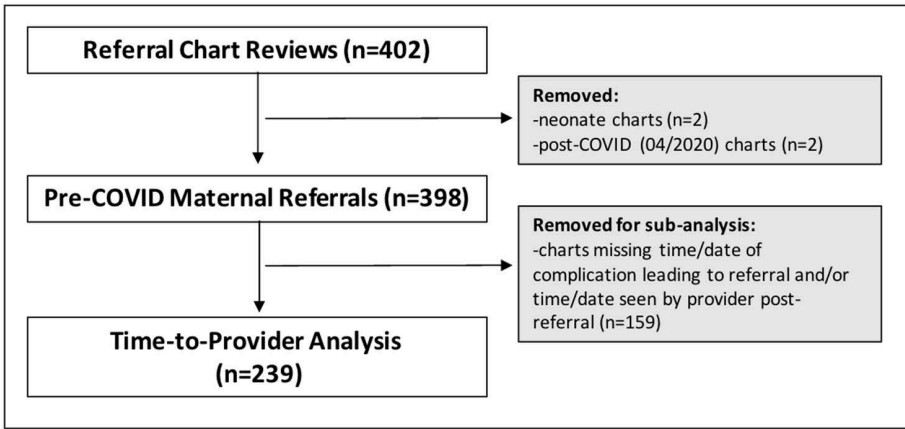

**Fig 2. Participant inclusion/exclusion flowchart.**

greater than 90% power, 402 charts were randomly chosen for review from the eligible patients. Of these charts, 398 were pre-COVID maternal charts and 239 included enough information to calculate a referral interval.

Chart reviews were performed by a Nurse-Midwife Research Coordinator, assisted by two trained Research Assistants (one Senior Nursing Officer and one Senior Medical Officer) who identified and reviewed eligible charts in September 2022. As clinical providers practicing locally these researchers had full chart access. A set list of non-identifiable patient-level variables was created in partnership between the GAIN research team and the Hospital leadership, with input from the Blantyre DHO (S1 Document). These were extracted from each chart to track each patient's journey from pre-referral to the outcome. Accordingly all other research team members did not have access to identifiable participant information during or after data collection.

Concurrently, qualitative data were drawn from providers (clinicians, nurses, and nurse-midwives) employed at both the referring facilities and at the Hospital. Provider perspectives were contextualized with interviews that included postpartum patients who were referred from a participating facility to the Hospital. Providers were approached by GAIN study team members and invited to participate. Postpartum patients who had undergone emergency referral to the Hospital were also approached by GAIN study team members and invited to participate during their one-week postnatal visit at their referring facility. Additionally, posters in English and Chichewa (local language) were posted across all participating facilities directing eligible individuals to the opportunity to participate in the study. All study participants who agreed to be interviewed provided written consent prior to data collection. All interview participants all were considered consenting adults adhering with Malawian and U.S. rules and regulations. Permission to decline participation without negative ramifications was emphasized for both groups.

Semi-structured qualitative focus groups and interviews were conducted in English and Chichewa by the Malawian Research Coordinator between February and June 2022. Recruitment for focus groups and interviews occurred during the same period on differing days—eligible participants who were available on the day each method occurred were invited. We used both in-depth interviews and focus-group discussions to enable us to capture individual experiences and communal perceptions around care respectively. Data collection was guided by open-ended questions and probes aimed at understanding the referral process including its timeliness, costs, drivers, bottlenecks, and participant recommendations for improvement (S2 Document). Three focus groups (two with providers, one with patients) of three to seven people lasted an average of 48 minutes while each of the 18 interviews (seven with patients, 11 with providers) lasted an average of 15 minutes. Both were audio-recorded, and field notes were taken by GAIN staff. All participants were compensated the local equivalent of 10USD.

## Data analysis

Quantitative data were entered into REDCap and transferred to RStudio for analysis [27]. Descriptive analyses were first conducted to summarize overall, and facility-level, patient characteristics and variables associated with the referral process. Desiring to identify opportunities for intervention along the referral care continuum, subsequent bivariate Kruskal Wallis and Fisher's exact tests were used to assess factors associated with maternal care and outcomes. Specifically, we assessed referral care variables (i.e., distance from the Hospital and total referral time) and maternal care variables (i.e., procedures completed and complications) by both arrival condition (stable versus critical) and maternal outcome (discharged/absconded versus death). Concordance between pre-referral and Hospital admission diagnoses was explored. Finally, bivariate tests also compared patient demographics and care variables for those in the full sample and sub-analyses to identify potentially significant differences that could affect the interpretation. For bivariate tests, a $p$-value of 0.05 or less was considered statistically significant.

Following the completion of qualitative interviews and focus groups, the audio files were translated and transcribed. All transcripts were reviewed, coded, and thematically analyzed by two Malawian nurse-midwife researchers and two American researchers using Microsoft Excel. Data were reduced using a matrix. A grounded theory approach was then used to

iteratively identify, apply, and categorize inductive codes into emergent themes across and between the practitioner and patient cohorts.

While qualitative data are essential to understanding nuances of the human experience in health systems like the referral care continuum, it is important to ensure their reliability. To ensure the trustworthiness of the qualitative data, interview and focus group facilitators collaborated with other members of the study team to review data and explore biases after collection. Having been engaged in research for many years in the district, the study team's longstanding relationship with clinical providers at the Hospital enhanced credibility of the data collected and transferability during the sampling process. Additionally, detailed documentation of decision-making processes contributed to dependability in data analysis.

Following initial analysis, quantitative and qualitative findings were compared together to assess congruency in trends. To gain a more comprehensive understanding of the barriers and facilitators throughout the referral process, triangulation was used to identify convergent, complementary, and/or contradictory results [28]. We explore this further in the discussion section. It was during the triangulation phase that our thematic findings were found to align with the fundamental components of health systems strengthening previously outlined by non-governmental organizations (NGOs) such as Partners in Health (PIH), a global organization committed to high-quality health care for vulnerable groups who operate in Malawi [29]. These components include "Five S's": staff, stuff, space, systems, and social support.

### Ethical considerations

The study was approved by the Hospital research committee, the National Health Sciences Research Committee (NHSRC) (#21/08/2767), and UCSF Human Research Protection Program (HRPP) (19-29585).

### Researcher characteristics, reflexivity, and word choice

This study was designed and conducted by a binational team, most of the members of which are nurses or nurse-midwives. Multidisciplinary decisionmakers and providers from the Ministry of Health (MoH), the Hospital, and referring facilities were engaged and involved since the inception of the study. All data collection was conducted by Malawian team members. Data analysis was conducted by a subset of members with qualitative, quantitative, and mixed methods research expertise and many years working in close collaboration with low-income settings in sub-Saharan Africa. Mindful of the influence researchers who are not from Malawi could have in the research process, analysis and interpretation were conducted iteratively, and routinely informed by Malawian team members. Finally, we have purposefully chosen to use the word "woman/women" and/or "patient/patients" throughout to describe the patient population of mothers and birthing persons in our study, aligning with local preferences.

## Results

### Quantitative findings

Among the 398 patients, about half (53.26%) were between 18 and 24 years-old and two-thirds (67.84%) were referred from a facility less than 10km from the Hospital (Table 1). Most women were referred during the intrapartum period (77.14%), admitted to the labor ward (94.22%), in stable condition upon arrival (70.35%), and discharged alive (93.47%). Most women presented one pre-referral condition (90.45%)—most commonly prolonged/obstructed labor followed by pre/eclampsia and other diagnoses. Fisher's Exact Tests revealed no significant differences between the maternal referral characteristic variables of the full sample of pre-COVID maternal referrals and the subset for whom referral interval could be calculated used in the sub-analyses.

Notably, there was missingness across patients' pre-referral documentation. Of the 398 charts reviewed, 86.18% included the time a complication was identified. Key pre-referral vital statistics also varied: 64.07% included maternal blood pressure (BP); 58.29% included maternal pulse; 20.60% included maternal temperature; and 14.32% included maternal respirations. Further, data on the referral process was sparse: 21.36% included the time transport was called;

**Table 1. Maternal referral characteristics for the full sample and the sample use for sub-analysis.**

| | All Maternal Referrals (N = 398) | Maternal Referrals for whom a referral interval could be calculated (n = 239) | *p-value* |
|---|---|---|---|
| **Age** | | | |
| Mean Age (SD) | 24.29 (20.0 - 28.0) | 24.11 (19.50 - 28.0) | |
| 18 and under | 21 (5.28%) | 14 (5.86%) | 0.984 |
| 18 to 24 years old | 212 (53.26%) | 128 (53.56%) | |
| 25 to 29 years old | 85 (21.36%) | 53 (22.18%) | |
| 30 to 34 years old | 57 (14.32%) | 32 (13.39%) | |
| 35 and above | 23 (5.78%) | 12 (5.02%) | |
| **Referring Facility by Distance from the Hospital** | | | |
| < 10km from the Hospital | 270 (67.84%) | 163 (68.20%) | |
| *Facility A* | 39 (9.80%) | 27 (11.30%) | 0.905 |
| *Facility C* | 91 (22.86%) | 51 (21.34%) | |
| *Facility E* | 80 (20.10%) | 47 (19.67%) | |
| *Facility G* | 60 (15.08%) | 38 (15.90%) | |
| ≥ 10km from the Hospital | 127 (31.91%) | 76 (31.80%) | |
| *Facility F* | 48 (12.06%) | 30 (12.55%) | 0.926 |
| *Facility B* | 56 (14.07%) | 34 14.23%) | |
| *Facility D* | 23 (5.78%) | 12 (5.02%) | |
| Unknown | 1 (0.25%) | 0 (0.00%) | |
| **Referral Period** | | | |
| Antepartum | 45 (11.31%) | 17 (7.115%) | 0.317 |
| Intrapartum | 307 (77.14%) | 191 (79.91%) | |
| Postpartum | 41 (10.30%) | 29 (12.13%) | |
| Unknown | 5 (1.26%) | 2 (0.84%) | |
| **Admission Ward** | | | |
| Labour Ward | 375 (94.22%) | 234 (97.91%) | 0.107 |
| Antenatal Ward | 10 (2.51%) | 1 (0.42%) | |
| Other Ward[a] | 7 (1.76%) | 1 (0.42%) | |
| Unknown | 6 (1.51%) | 3 (1.26%) | |
| **Total Pre-referral Conditions** | | | |
| None | 8 (2.01%) | 2 (0.84%) | 0.455 |
| One | 360 (90.45%) | 222 (92.89%) | |
| Two or Three | 28 (7.04%) | 14 (5.86%) | |
| Unknown | 2 (0.50%) | 3 (1.26%) | |
| **Arrival Condition** | | | |
| Stable | 280 (70.35%) | 167 (69.87%) | 0.991 |
| Critical | 45 (11.31%) | 28 (11.72%) | |
| Unknown | 73 (18.34%) | 44 (18.41%) | |
| **Outcome** | | | |
| Discharged | 372 (93.47%) | 220 (92.05%) | 0.882 |
| Died | 10 (2.51%) | 8 (3.35%) | |
| Absconded | 5 (1.26%) | 4 (1.67%) | |
| Unknown | 11 (2.76%) | 7 (2.93%) | |

[a]Refers to any clinical ward that was not the labor or antenatal ward.

36.43% included the mode of transport; and 2.26% included the time of hospital arrival. Similarly, accompaniment of the referral was largely unrecorded. For example, "escort present" was recorded as "No" (6.78%) or "Not Listed" (93.22%) for all women, and no chart listed a consultation with the flying squad.

**Findings from All maternal referrals.** Analyses of the full study cohort revealed key referral characteristic—aspects of a patient's health or care during the referral period—differences by maternal condition upon arrival to the Hospital and by final maternal outcome. Compared to those who arrived in stable condition, those arriving in critical condition were significantly ($p < 0.01$) more likely to receive IV fluids, medications, or two or more hospital procedures (including catheterization, cannulation, etc.) post-referral (Table 2). In addition, those who arrived in critical condition were significantly more likely to experience complications during their stay and to experience death. While not statistically significant, those who arrived in critical condition also had a longer referral interval on average.

Compared to those who lived, women who died were significantly more likely to have been referred from a facility more than 10 km away, to have arrived in critical condition, and 100% experienced complications during their stay (Table 3). Further analysis, using Spearman's Test, of the relationships between these variables demonstrated that while facility distance and the occurrence of complications were not related, there was moderate correlation between facility distance and arrival condition (rho: 0.220, $p = 0.002$) [30]. We also identified strong positive correlation between facility and distance (rho: 0.682, $p < 0.001$). Additionally, all of those who died had at least one pre-referral diagnoses whereas seven discharged/absconded patients had none.

A descriptive analysis revealed some concordance between diagnoses from the referring facility versus upon admission at the Hospital, with notable exceptions among fetal distress, premature labor, and prolonged/obstructed labor (Table 4). The reports of ruptured uterus were the only completely concordant diagnosis among those analyzed. Further exploration of women with the most common pre-referral diagnosis of prolonged/obstructed labor (n = 129) revealed that 99 arrived in stable condition and five in critical condition—the arrival condition was not recorded for 25 of these women. Additionally, 76 births occurred via cesarean section (one cesarean and hysterectomy combination), three via vacuum extraction, and four were forceps-assisted. Though no APGAR scores were recorded for any of the infants born to women with a pre-referral diagnosis of prolonged/obstructed labor, three were indicated to have been born asphyxiated. Further, four fresh stillbirths and one neonatal death were recorded among the women.

Review of pre-referral clinical procedures revealed that, apart from the routine uterotonics provided to women after delivery, oxytocin infused in Normal Saline/ Ringer's Lactate (NS/RL) was the most common intervention received for women with a pre-referral diagnosis of postpartum hemorrhage while antihypertensives and anticonvulsants were the most common among patients with pre-/eclampsia. Additionally, fatal maternal complications included postpartum hemorrhage, hypovolemic shock, and cardiac arrest among other conditions. In several instances, a pre-referral intervention was documented as not done due to stock-outs in drugs or supplies. Further exploration of reported stock-outs found that for 80 women, medications or supplies were reported to be unavailable during the pre-referral period. Of these, supplies for catheterization (43.75%) were most reported as missing. There were also 20 instances in which a blood pressure machine or batteries for it were not available and eight instances when urine dipsticks were out. Finally, there were five instances in which needed IV fluids were unavailable and five instances when methyldopa was missing, and several other less frequently mentioned stock-outs of other items.

**Findings from those for whom a referral interval could be calculated.** The Time-To-Provider sub-analysis found that referral interval varied across circumstances and environments (Table 5). The average referral interval varied by referring facility with an average time from complication to provider of 5.05 (7.22) from Facility D, the furthest facility, to 1.75 hours (SD: 0.80) from Facility G, the closest ($p = 0.01$). Compared to those referred during antepartum or postpartum periods, women referred during intrapartum experienced a shorter referral interval ($p < 0.01$). Additionally, time was significantly shorter for women experiencing prolonged/obstructed labor ($p = 0.03$) compared to those without a prolonged/obstructed labor diagnosis (S1 Table).

**Table 2. Key maternal and intervention characteristics for all maternal referrals by condition upon arrival at the hospital.**

| | Condition Upon Arrival[a] | | |
| --- | --- | --- | --- |
| | **Stable**<br>**(n = 280, 70.30%)** | **Critical**<br>**(n = 45, 11.50%)** | *p-value* |
| **Average Age (SD)** | 23.9 (5.57) | 24.6 (6.64) | 0.548 |
| **Average Hrs to Provider (SD)** | 2.98 (3.87) | 4.22 (6.22) | 0.354 |
| **Total Pre-referral Diagnoses** | | | |
| None | 5 (1.8%) | 3 (6.7%) | 0.274 |
| 1 | 252 (90.0%) | 39 (86.7%) | |
| 2 | 22 (7.8%) | 3 (6.7%) | |
| 3 | 1 (0.4%) | 0 (0.0%) | |
| **Referring Facility Distance to the Hospital** | | | |
| <10 km | 192 (68.6%) | 24 (53.3%) | 0.060 |
| ≥10 km | 87 (31.1%) | 21 (46.7%) | |
| **Total Diagnoses upon Admission** | | | |
| None | 12 (4.3%) | 5 (11.1%) | <0.001 |
| 1 | 241 (86.1%) | 29 (64.4%) | |
| 2 | 27 (9.6%) | 9 (20.0%) | |
| 3 | 0 (0.0%) | 2 (4.4%) | |
| **Total Hospital Procedures** | | | |
| None | 171 (61.1%) | 17 (37.8%) | <0.001 |
| 1 | 69 (24.9%) | 14 (31.1%) | |
| 2 | 40 (14.3%) | 12 (26.7%) | |
| 3 | 0 (0.0%) | 2 (4.4%) | |
| **Total IV Fluids Given[*]** | | | |
| None | 206 (73.6%) | 25 (55.6%) | 0.004 |
| 1 | 66 (23.6%) | 14 (31.1%) | |
| 2 | 7 (2.5%) | 6 (13.3%) | |
| 3 | 1 (0.4%) | 0 (0.0%) | |
| **Total Medications Given[**]** | | | |
| None | 222 (79.3%) | 23 (51.1%) | <0.001 |
| 1 | 47 (16.8%) | 18 (40.0%) | |
| 2 | 8 (2.9%) | 3 (6.7%) | |
| 3 | 2 (0.7%) | 1 (2.2%) | |
| 4 | 1 (0.4%) | 0 (0.0%) | |
| **Complications During Stay** | | | |
| Yes | 69 (24.6%) | 27 (60.0%) | <0.001 |
| No | 211 (75.4%) | 18 (40.0%) | |
| **Maternal Outcome** | | | |
| Discharged | 268 (95.7%) | 33 (73.3%) | <0.001 |
| Died | 3 (1.1%) | 7 (15.6%) | |
| Absconded | 3 (1.1%) | 2 (4.4%) | |

[a]*Excludes persons for whom arrival condition was unknown/not recorded.*

[*]*IV fluids included Normal Saline, Ringers Lactate, Dextros 5%, Dextros 50%, Haemacele, Oxytocin in NS/LR, and Other.*

[**]*Medications included Anticonvulsants, Antihypertensives, Antibiotics, Antimalarials, Uterotonics, Tocolytics, Corticosteroids, Analgesics, Laxatives, and Other including Benzylpenicillin and Magnesium Sulfate.*

**Table 3. Key maternal and intervention characteristics for all maternal referrals by final patient outcome.**

| | Maternal Patient Outcome[a] | | |
|---|---|---|---|
| | Discharged/Absconded (n = 377, 94.72%) | Died (n = 10, 2.51%) | *p-value* |
| **Average Age (SD)** | 24.3 (5.63) | 27.8 (7.76) | 0.154 |
| **Average Hrs to Provider (SD)** | 3.17 (4.14) | 2.44 (1.78) | 0.866 |
| **Total Pre-referral Diagnoses** | | | |
| None | 7 (1.9%) | 0 (0%) | 0.032 |
| 1 | 344 (91.2%) | 8 (80.0%) | |
| 2 | 26 (6.9%) | 1 (10.0%) | |
| 3 | 0 (0.0%) | 1 (10.0%) | |
| **Referring Facility Distance to the Hospital** | | | |
| <10 km | 260 (69.0%) | 3 (30.0%) | 0.014 |
| ≥10 km | 116 (30.8%) | 7 (70.0%) | |
| **Total Diagnoses upon Admission** | | | |
| None | 25 (6.6%) | 0 (0.0%) | <0.001 |
| 1 | 310 (82.2%) | 5 (50.0%) | |
| 2 | 42 (11.1%) | 3 (30.0%) | |
| 3 | 0 (0.0%) | 2 (20.0%) | |
| **Total Hospital Procedures** | | | |
| None | 212 (56.2%) | 1 (10.0%) | 0.012 |
| 1 | 103 (27.3%) | 5 (50.0%) | |
| 2 | 58 (15.4%) | 4 (40.0%) | |
| 3 | 4 (1.1%) | 0 (0.0%) | |
| **Total IV Fluids Given** | | | |
| None | 257 (68.2%) | 4 (40.0%) | 0.07 |
| 1 | 99 (26.3%) | 4 (40.0%) | |
| 2 | 20 (5.3%) | 2 (20.0%) | |
| 3 | 1 (0.3%) | 0 (0.0%) | |
| **Total Medications Given** | | | |
| None | 286 (75.6%) | 6 (60.0%) | 0.114 |
| 1 | 73 (19.4%) | 3 (30.0%) | |
| 2 | 15 (4.0%) | 0 (0.0%) | |
| 3 | 3 (0.8%) | 1 (10.0%) | |
| 4 | 1 (0.3%) | 0 (0.0%) | |
| **Complications During Stay** | | | |
| Yes | 266 (70.6%) | 10 (100%) | <0.001 |
| No | 111 (29.4%) | 0 (0.0%) | |
| **Condition Upon the Hospital Arrival** | | | |
| Stable | 271 (71.9%) | 3 (30.0%) | <0.001 |
| Critical | 35 (9.3%) | 7 (70.0%) | |

[a]*Excludes persons for whom outcome was unknown/not recorded.*

Further exploration of those with relatively short (<2 hours), moderate (2–4 hours), and long (≥4 hours) referral intervals revealed additional associations (Table 6). Compared to those with shorter wait times between complication to being seen by a provider, those whose referrals took four or more hours were significantly more likely to be from a referring facility

**Table 4. Maternal diagnosis dis/concordance for all maternal referrals between pre-referral and admission to the hospital.**

| Diagnosis | Total Diagnoses | Diagnosed Pre-referral Only | Diagnosed on Admission Only | Diagnosed Both Times[a] |
|---|---|---|---|---|
| Antepartum hemorrhage | 16 | 6 (37.50%) | 2 (12.50%) | 8 (50.00%) |
| Fetal distress | 44 | 29 (65.91%) | 9 (20.45%) | 6 (13.64%) |
| Postpartum hemorrhage | 25 | 10 (40.00%) | 6 (24.00%) | 9 (36.00%) |
| Pre-/eclampsia | 87 | 33 (37.93%) | 7 (8.05%) | 47 (54.02%) |
| Premature labor | 14 | 8 (57.14%) | 1 (7.14%) | 5 (35.71%) |
| Retained placenta | 12 | 1 (8.33%) | 0 (0.00%) | 11 (91.67%) |
| Ruptured uterus | 2 | 0 (0.00%) | 0 (0.00%) | 2 (100.00%) |
| Sepsis | 3 | 1 (33.33%) | 2 (66.67%) | 0 (0.00%) |
| Prolonged/ obstructed labor | 131 | 84 (64.12%) | 2 (1.53%) | 45 (34.35%) |
| Asphyxia | 1 | 1 (100.00%) | 0 (0.00%) | 0 (0.00%) |
| None Listed | 35 | 7 (20.00%) | 27 (77.14%) | 1 (2.86%) |

[a]Diagnosis concordance shading: <50% (red); 50% - 75% (yellow); >75% (green).

>10km away (p = 0.06), to be referred during the antepartum period (p < 0.01), and to experience a complication post-referral at the Hospital (p = 0.02).

## Qualitative findings

Analysis of in-depth interview and focus group transcripts uncovered three emergent and interrelated themes explaining the facilitators and barriers to timely, quality obstetric referrals: 1) systemic and structural challenges; 2) inconsistent inter- and intra-facility communication; and 3) social and provider influences on maternal expectations and beliefs. These factors were described to influence care across the referral continuum and to shape maternal perspectives, experiences, and outcomes.

**Systemic and structural challenges.** Both providers and patients repeatedly described systemic and structural inefficiencies including staffing shortages at both the referring facilities and QECH. At referring facilities, patients expressed concerns that insufficient staffing may result in referrals without clinical indication toward achieving manageable caseloads. Long wait times and worsened care quality were described to result from overburdened providers juggling many patients at once. One provider explained, "*I wanted to insert an IV line but at this time, I was the only nurse but luckily a clinician came. When the clinician came, we helped each other until we had started the IV line*" (Provider Focus Group Participant, Referral Facility). The high patient to provider ratio is not uncommon across all facilities in the district and contribute to care delays. Participants suggested that long delays were particularly common upon admission to the labor ward at the referral hospital where a single midwife may be the sole staff managing referrals and the operating theatre where specialists such as anesthetists are in high demand. One participant explained, "*An anesthetist is like gold, aah for them to be available? Because they are overwhelmed just like us.*" (Provider Focus Group Participant, the Hospital). Many health workers also said the staffing shortage often led to them feeling exhausted and overwhelmed, and potentially contributing to poor attitudes towards patients.

A shortage of essential medical resources at referring sites and inadequate physical space to receive referrals at the Hospital further impacted the timeliness and quality of care. Patients perceived that referrals could potentially be reduced if referring sites consistently had what they needed to provide comprehensive care. Providers expressed a similar frustration about how a lack of resources limited their ability to provide the care they wished they could. One participant explained,

"*There is a lot, talk of gloves, how would you examine a patient without gloves? No way! Even electricity. sometimes there is a blackout so someone with [postpartum hemorrhage] comes but you are using a candle will that work?*" (Provider Interview Participant, Referral Facility).

**Table 5. Referral interval (hours) by maternal and referral characteristics among those for whom a referral interval could be calculated.**

| | Referral Interval in Hours | | |
|---|---|---|---|
| | Median [Range] | Mean (SD) | p-value |
| **Arrival Condition** | | | |
| Stable | 2.00 [0.417, 29.7] | 2.96 (3.84) | 0.334 |
| Critical | 2.13 [0.583, 25.5] | 4.22 (6.22) | |
| **Referral Period** | | | |
| Antepartum | 3.75 [0.917, 10.3] | 3.69 (2.39) | 0.008 |
| Intrapartum | 1.92 [0.417, 29.7] | 2.92 (3.99) | |
| Postpartum | 2.47 [0.583, 25.5] | 3.56 (4.52) | |
| **Mode of transport** | | | |
| Ambulance | 1.96 [0.417, 29.7] | 2.69 (3.49) | 0.770 |
| Personal Vehicle | 2.17 [0.500, 10.3] | 3.92 (3.71) | |
| Hired Vehicle | 1.88 [0.583, 4.75] | 2.08 (1.16) | |
| **Maternal Outcome** | | | |
| Discharged | 2.00 [0.417, 29.7] | 3.15 (4.15) | 0.756 |
| Absconded | 2.04 [0.917, 6.50] | 3.02 (1.69) | |
| Died | 3.00 [1.33, 4.75] | 2.44 (1.78) | |
| **Mode of delivery** | | | |
| Vaginal | 2.00 [0.450, 29.7] | 3.57 (4.66) | 0.170 |
| Cesarean | 1.92 [0.417, 27.0] | 2.51 (3.03) | |
| Vacuum Extraction | 2.82 [1.08, 7.67] | 3.46 (2.51) | |
| Breech | 1.18 [0.950, 1.42] | 1.18(0.33) | |
| Forceps-assisted | 1.46 [0.667, 2.25] | 1.46 (1.12) | |
| **Complications during stay** | | | |
| Yes | 2.25 [0.500, 25.5] | 3.22 (3.58) | 0.167 |
| No | 2.00 [0.417, 29.7] | 3.04 (4.17) | |
| **Facility** | | | |
| Facility A | 2.55 [0.833, 6.50] | 3.06 (1.47) | 0.007 |
| Facility C | 1.92 [0.450, 25.3] | 3.16 (4.25) | |
| Facility E | 1.96 [0.417, 15.8] | 2.73 (3.00) | |
| Facility F | 2.08 [0.667, 25.5] | 4.11 (5.45) | |
| Facility G | 1.60 [0.717, 3.58] | 1.75 (0.80) | |
| Facility B | 2.17 [0.500, 29.7] | 3.48 (4.99) | |
| Facility D | 2.45 [1.25, 27.0] | 5.05 (7.22) | |
| **Distance from the Hospital** | | | |
| <10km | 1.92 [0.417, 25.3] | 2.68 (2.99) | 0.061 |
| >10km | 2.08 [0.500, 29.7] | 3.98 (5.52) | |
| **Day of Referral[b]** | | | |
| Weekday | 2.00 [0.500, 29.70] | 3.29 (4.44) | 0.778 |
| Weekend day | 2.00 [0.41, 10.30] | 2.43 (1.74) | |
| **Time of Day Related to Clinical Shifts[c]** | | | |
| Daytime | 2.00 [0.417, 29.7] | 2.87 (3.53) | 0.637 |
| Overnight | 2.00 [0.58, 27.0] | 3.55 (4.83) | |
| Shift change | 1.92 [0.50, 3.33] | 1.84 (1.02) | |

[a]Note that data were not available on whether breech births were spontaneous or assisted and were therefore analyzed separately.

[b]"Weekday" refers to Monday through Friday; "Weekend day" refers to Saturday and Sunday.

[c]"Daytime" refers to 8 am to 4 pm local time; "Overnight" refers to 5 pm to 7 am; "Shift change" refers to 7 am to 8 am and 4 pm to 5 pm.

**Table 6. Key maternal characteristics and outcomes among those for whom a referral interval could be calculated by referral interval category.**

| | Total Referral Interval in Hours | | | |
| --- | --- | --- | --- | --- |
| | <2 hrs (N = 110) | 2 to 4 hrs (N = 87) | >4 hrs (N = 39) | p-value |
| **Mean Age (SD)** | 24.9 (5.66) | 23.7 (5.90) | 22.8 (4.68) | 0.095 |
| **Distance to the Hospital** | | | | |
| <10km | 82 (50.93%) | 58 (36.02%) | 21 (13.04%) | 0.058 |
| ≥10km | 28 (37.33%) | 29 (38.67%) | 18 (24.00%) | |
| **Total Pre-Referral Diagnoses** | | | | |
| None | 0 (0.00%) | 2 (100.00%) | 0 (0.00%) | 0.225 |
| 1 | 101 (46.12%) | 79 (36.07%) | 39 (17.81%) | |
| 2+ | 8 (57.14%) | 6 (42.86%) | 0 (0.00%) | |
| **Referral Period** | | | | |
| Antepartum | 5 (29.41%) | 5 (29.41%) | 7 (41.18%) | 0.009 |
| Intrapartum | 95 (50.53%) | 70 (37.23%) | 23 (12.23%) | |
| Postpartum | 9 (31.03%) | 12 (41.38%) | 8 (27.59%) | |
| **Arrival Condition** | | | | |
| Stable | 79 (45.12%) | 61 (37.20%) | 24 (14.63%) | 0.163 |
| Critical | 13 (46.43%) | 7 (25.00%) | 8 (28.57%) | |
| **Total Procedures** | | | | |
| None | 50 (45.45%) | 38 (34.55%) | 22 (20.00%) | 0.408 |
| 1 | 32 (45.07%) | 27 (38.03%) | 12 (16.90%) | |
| 2 | 28 (51.85%) | 21 (38.89%) | 5 (9.26%) | |
| 3 | 0 (0.00%) | 1 (100.00%) | 0 (0.00%) | |
| **Complications During Stay** | | | | |
| No | 82 (49.10%) | 65 (38.92%) | 20 (11.98%) | 0.019 |
| Yes | 28 (40.58%) | 22 (31.88%) | 19 (27.54%) | |
| **Outcome** | | | | |
| Discharged | 102 (47.00%) | 79 (36.41%) | 36 (16.59%) | 0.372 |
| Died | 4 (50.00%) | 3 (37.50%) | 1 (12.50%) | |
| Absconded | 2 (50.00%) | 0 (0.00%) | 2 (50.00%) | |

Participants similarly recognized the limited capacity of the Hospital to receive all district referrals. They described inadequate physical space to effectively conduct admissions, monitor laboring women, and conduct surgical operations. One participant shared, "*We have a lot of deliveries in the delivery room and we are overwhelmed and in the delivery area we have only 24 beds, we have the whole Blantyre against us*" (Provider Focus Group Participant, Referral Facility).

Finally, most participants cited a lack of reliable transportation as a critical barrier to the referral process. Participants perceived an insufficient supply of ambulances, resulting in long wait times, lengthier referral intervals, and unavailable transportation altogether. These challenges were anecdotally described to result in poorer health outcomes for women and their infants including lower Apgar scores—the latter of which were not recorded prohibiting quantitative confirmation of this finding. Transportation issues were compounded by fuel shortages which were described to cause additional delays. While some patients had means to acquire alternate transportation, others were left waiting. One participant explained,

"*Mostly they say there is no fuel for the ambulance to come, sometimes there is only one ambulance and is very far away so for a patient with [postpartum hemorrhage], it is very difficult to wait thinking they will come in time.*" (Provider Interview Participant, Referral Facility).

**Inconsistent inter- and intra-facility communication.** Participants discussed that referral care was often delayed by breakdowns in communication including challenges reaching the 'Flying Squad'—a team of on-call providers available for virtual consultation for patients requiring referral. While the Flying Squad was described to be historically beneficial to consult, during the time of the study some calls went unanswered, and some Squad members were described to lack essential maternal health knowledge. Communication between the referring facilities and the referral hospital was also described to have mixed success and breakdowns were described to result in worse care. One provider explained,

"*I think it [a lack of communication] affects the quality of care because, one, we have to start preparing when the baby is actually there and I think that time could have been shortened if they had told us they are bringing a baby with such a condition, and the baby will need this and that, we are going to prepare a cot and equipment the baby will need*" (Provider Interview Participant, the Hospital).

Another provider participant shared that the lack of documentation (written communication) created further challenges "*[…] because it happens that people come without a partograph, but you are told this is prolonged labour now you do not know where to start from*" (Provider Interview Participant, the Hospital). Dialogue via WhatsApp sometimes facilitated a smooth handoff though was not routinely done and written documentation of pre-referral management was often incomplete.

**Social and provider influences on maternal expectations and beliefs.** Most patients described negative feelings about being referred to the Hospital which were informed by personal experiences or by stories they had heard from others. Community dialogue surrounding perceived quality of care sometimes influenced patient expectations and participants perceived they negatively impacted patient behavior and wellbeing. One provider reported a patient who ultimately absconded due to their frustrations with delayed referral care while one patient expressed regret about her transfer due to the treatment delays that she experienced. One patient shared,

"*…So I was worried to say ah ah, with the way people talk about [the Hospital] saying there are a lot of people and sometimes it happens that others are in the corridors, and they deliver on their own so how will it turn out for me?*" (Patient Focus Group Participant, Referral Facility).

On the other hand, women who understood the reasons for their referral or anticipated it ahead of time were more understanding and comfortable with the process. One participant shared, "*I took it normally since I was told beforehand that I would not deliver here since I have a caesarean in my previous pregnancy*" (Patient Interview Participant, Referral Facility). In general, participants underscored the need for both patient-provider and provider-provider communication.

## Discussion

Our findings confirmed that pre-referral challenges were associated with worse maternal morbidity and mortality while shedding light upon opportunities for intervention across the referral care spectrum [6]. Importantly, our mixed-methods approach not only revealed challenges across the referral process, but also highlighted targeted opportunities to improve structural factors and communication from beginning to end. This expands upon prior findings suggesting multifaceted referral challenges in Malawi [7,8].

Unsurprisingly, findings showed that women who arrived to the Hospital in critical condition fared worse than those in stable condition. This highlights the need to train staff at primary care facilities on early identification and timely referral of potentially life-threatening complications. We also found that those who died were more likely to have traveled further to receive tertiary care at the Hospital, and to have experienced a complication post-referral. While few previous studies

focus on transportation during obstetric emergencies in low-resource settings, our findings align with reviews that suggest it is a major barrier to timely care which influences worse downstream outcomes [31,32]. Additional research is needed to better understand the relationship between distance to care and outcomes, in addition to the drivers of post-referral complications.

Qualitative findings from both patients and providers underscored the prevalence of transportation limitations, communication breakdowns between facilities, shortages in human and material resources, and limited infrastructure. These concerns aligned with the referral delays and adverse health outcomes that we had identified and contextualized as possible contributing factors. Our findings also align with prior research in low-resource settings suggesting that a longer time waiting for transport and/or spent traveling based on distance and terrain resulted in a worse patient experience in addition to increasing maternal morbidity and mortality [32]. This work further underscores previously identified gaps in maternal referral care which increases both provider stress and patient dissatisfaction [33].

## Triangulated findings

Triangulation of our quantitative and qualitative findings highlighted convergent themes related to the referral process. This included reviews of patient charts with providers to explore potential explanations for adverse outcomes as well as strategies that could reduce them. Our findings aligned with the fundamental components of health systems strengthening, the "Five S's": staff, stuff, space, systems, and social support. Similar to prior findings, we found that the absence of any one of these components results in a weaker referral care system [29].

**Impact of staffing.** An optimal system would include an adequate number of well-trained providers and other staff to offer timely care across the referral process. Unfortunately, our study participants noted that access to providers with specialty skills was limited which confirms previous findings around the impact of inadequate staff training, deployment, and retention [34]. While not explicit in our findings, we suspect that staffing shortages contributed to the missingness in data on patient referral paperwork that was apparent across patient charts in this study. Understandably, providers were likely focused on direct patient care and documentation was not prioritized. This was particularly apparent in the admission process at the Hospital with scant documentation of patient arrival time and none specifying whether the patient was accompanied by someone during their transfer. While these variables are less clinically significant than patient vital signs–which also were subject to missingness–they are important for those seeking to improve the entire process.

Unlike other research in similar contexts however, we did not find significantly different referral intervals by day of the week or by time of the day, even when accounting for shift changes [35]. We hypothesize this could be as a result of the structural challenges that worsen transportation, staffing, and space, regardless of the time or day.

**Impact of 'stuff'.** Well-functioning referral systems should ideally have adequate and appropriate resources to serve those requiring a higher level of care. Obstetric referrals, in particular, demand timely access to safe transportation, reliable access to well-functioning health resources including blood-pressure cuffs, and surgical resources for emergency cesarean section deliveries if needed. In our study, however, facility- and district-level stockouts of critical supplies prevented interventions for at least 80 patients. Qualitatively, care was also described to be routinely inhibited by a lack of medical supplies including gloves and by unpredictable electricity outages. Gaps in needed supplies were described to add stress for providers and increase risk of worse outcomes for patients. These challenges are a known source of worsened obstetric care and outcomes across sub-Saharan Africa [25,36–38].

**Impact of space.** Ideally, referral hospitals should have sufficient, well equipped, safe space to accommodate the load of regular and transferred patients. Unfortunately, providers described a lack of physical space both in terms of the number of delivery rooms and surgical theatres. This aligns with prior research showing restricted obstetric space lengthens care delays, and inevitably worsens outcomes, for referred patients [12,34]. Among the referring facilities, only one (Facility E) had the capacity to perform elective cesarean section deliveries on specific days of each month, further limited by funding constraints. While patient charts from Facility E were comparable with other facilities, it is possible

that with adequate support they could prevent some of the referrals that other sites would not be able to—reducing the backlog of patients awaiting surgical intervention at the Hospital. The lack of space at the Hospital was also highlighted as a critical concern for patients. Many had heard stories of people delivering on their own in crowded corridors. This resulted in providers describing patients who they perceived to resist or even flee care if a transfer was required, resulting in further delays and likely worse outcomes.

**Impact of systems.** Strong referral systems require appropriate leadership, governance, and financing, as well as robust transfers of information. It is therefore significant that the facilities, Hospital, and MoH are engaged in this research to identify areas for improvement including improved communication across the referral continuum. Patients who perceived their transfer to be unexpected or who were not given clear reasons from the provider were most hesitant about the referral process. This finding underscores the importance of patient-provider communication which may have been overlooked in some cases due to medical urgency or the understaffing described above. Facility distance from the Hospital, for instance, is what most impacted referral time and those whose referrals took at least four hours were more likely to experience an additional complication post-referral. This highlights inadequate transportation systems, particularly for facilities far from tertiary care. The insufficient supply of ambulances and unreliable access to fuel heightened fear among referring providers and patients waiting to be referred. This particular delay was also disproportionately faced by women without the financial means to acquire appropriate, alternate transportation. Our results confirm the previously noted need for community-informed transport interventions and infrastructure—as well as improved health financing to sustain solutions [12,25].

Our findings also point to a lack of inter- and intra-facility communication which negatively impacted patients and providers alike. Providers from referring facilities described challenges reaching the Flying Squad as well as staff at the Hospital. These findings reiterate prior findings which suggest that upstream outages, such as nonfunctioning radios, likely contributed to communication breakdowns during referrals [8]. On the other end of referrals, providers at the Hospital described the stress of receiving a patient without their partograph or ones that were incomplete. This resulted in additional time re-evaluating transfers which is burdensome for the already understaffed hospital and potentially worrisome for patients. Even when pre-referral information was available, we found discordance between diagnoses made at the referring facility and those at the Hospital. These realities along with differing criteria for diagnoses such as fetal distress, ultimately inform additional care delays.

**Impact of social support.** Excellent social support for referrals entails the provision of basic resources and trustworthy systems to ensure holistic care for both patients and their support system. Instead, as seen elsewhere in prior research, we found that patients were aware and concerned about the added risks of referral in an imperfect system [39]. In our study, patients who had not yet experienced referral themselves still expressed negative perspectives towards it, highlighting the influence of social and familial narratives surrounding care experiences. Like providers, patients and their support persons were also frustrated when there were delays with their transfer once it was determined to be necessary. Altogether, our findings confirm that an empathetic approach is needed to overcome socially embedded mistrust identified by participants in our study and others exploring obstetric referral care [25].

### Recommendations

With challenges across each component of referral care, a multi-pronged strategy will be necessary to improve the state of obstetric referrals and to reduce care gaps and delays. Our findings suggest that:

- Providers across referring health facilities need additional training and support to improve triaging, monitoring of labor, diagnosis precision, effective preventative care, and comprehensive pre-referral management.

- Transportation options should be increased and maintained by the government, particularly at sites further from tertiary care, to ensure quality and timely referral.

- Providers across the referral spectrum need consistent access to supplies to strengthen completeness of referral care documentation to ensure timely interventions.

- Admitting personnel and providers at referral hospitals need additional training and support to ensure accurate and comprehensive care for all transferred mothers.

A recognizable pattern across these referrals is the need for training and support of all referral staff. This aligns with recent research demonstrating that staff mentorship is critical for Malawian frontline nurses [40]. Relevant skills training and longitudinal mentorship in particular have demonstrated knowledge retention, competence, and confidence in the use of basic emergency skills which improve quality of care for patients like those in our study [41,42].

As part of the overall GAIN partnership with the facilities, the Hospital, and MoH in the district, these reccomendations and ways to make them actionable are currently being explored. The study team will continue with the piloting of an intervention to address each of the identified areas at all levels of the care continuum with results regularly disseminated to relevant stakeholders.

### Strengths and limitations

Our study applied multiple methods to understand the complex barriers to optimal referral care in Blantyre District, Malawi. Our findings have the potential to inform multi-level interventions toward strengthening the referral system and preventing maternal morbidity and mortality. This aligns with the WHO Vision on Quality of Care for Maternal and Newborn Health and addresses a key bottleneck of health service delivery identified in the Government of the Republic of Malawi Health Sector Strategic Plan III 2023–2030 [2,9]. The use of our binational team coupled with early and ongoing engagement with the Malawian MoH, the Hospital, and referring facilities also ensures short- and long-term action can be taken toward solutions.

Our most notable limitation was the lack of complete documentation of referral care which was shaped by the prior lack of standardized pre-referral documentation tools. Unlike comparable studies which could calculate post-referral hospital wait-time separate from transport time, our referral interval spanned from the time of complication at a referring facility to the time seen by provider [35]. Similarly, other research in comparable settings has suggested that poor management of patients during transit and a lack of professional escort for patients contributes to worse referral outcomes during obstetric emergencies [39]. While we attempted to capture consultations with the local escort team, there was no documentation of this. Further multivariate analyses of more robust data sets would also support the identification of potential predictive factors in the presence of confounders.

Finally, aligning with gaps in knowledge about facility readiness, our findings highlight barriers related to the third delay of the "three delays model" commonly used to understand maternal disparities [13,25]. The first delay relates to an individual's decision to seek care, the second delay focuses on their ability to reach a health care facility, and the third delay encompasses care that they did or did not receive. While our focus on the third delay is a strength in its contribution to literature, it must be acknowledged that our study only includes those who overcame the first two delays as they sought care and reached a referral facility—albeit many experienced delays with the latter. Prior findings in Malawi suggested that type three delays most contribute to maternal mortality but that all types of delays are frequent [12].

### Conclusions

Building the capacity of the referral system in Malawi is essential to ensuring the timely provision of emergency obstetric care and reducing maternal and neonatal morbidity and mortality. Our study confirmed the need for multifaceted solutions to referral care challenges in settings where resources are constrained. Persistent gaps in staffing, supplies, and space along with inadequate systems and social support have resulted in worse maternal care and outcomes. Women furthest from tertiary care facilities and those who experienced delayed referrals or care bore the brunt of this. Strategic investment in a well-functioning referral system should continue to be a local and global priority.

## Supporting information

**S1 Document.  Blantyre DHO referral analysis data collection form.**
(PDF)

**S2 Document.  Semi-structured qualitative interview guides for both clinicians and clients.**
(PDF)

**S1 Table.  Table of referral interval (hours) by pre-referral diagnoses.**
(PDF)

**S1 Checklist.  Inclusivity in global research.**
(DOCX)

## Author contributions

**Conceptualization:** Alden Blair, Kimberly Baltzell, Joyce Jere.

**Data curation:** Ashley Mitchell, Alden Blair, Oveka Mwanza.

**Formal analysis:** Ashley Mitchell, Luseshelo Simwinga, Miranda Rouse, Kimberly Baltzell, Richard Malirakwenda, Oveka Mwanza.

**Funding acquisition:** Kimberly Baltzell.

**Investigation:** Luseshelo Simwinga, Richard Malirakwenda, Oveka Mwanza.

**Methodology:** Ashley Mitchell, Alden Blair, Oveka Mwanza.

**Project administration:** Luseshelo Simwinga, Alden Blair, Miranda Rouse, Kimberly Baltzell, Richard Malirakwenda, Oveka Mwanza.

**Supervision:** Alden Blair, Oveka Mwanza.

**Visualization:** Ashley Mitchell.

**Writing – original draft:** Ashley Mitchell, Oveka Mwanza.

**Writing – review & editing:** Ashley Mitchell, Luseshelo Simwinga, Alden Blair, Miranda Rouse, Kimberly Baltzell, Richard Malirakwenda, Joyce Jere, Oveka Mwanza.

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
