## [Decision Letter · Decision Letter 0]

PGPH-D-24-00015

Characterizing the referral care continuum among complex obstetric patients in the Blantyre District of Malawi: A mixed methods study

Dear Dr. Mitchell,

Thank you for submitting your manuscript to PLOS Global Public Health. After careful consideration, we feel that it has merit but does not fully meet PLOS Global Public Health’s publication criteria as it currently stands. Therefore, we invite you to submit a revised version of the manuscript that addresses the points raised during the review process.

We look forward to receiving your revised manuscript.

Kind regards,

Hannah Tappis, DrPH, MPH

Academic Editor

Journal Requirements:

1. The resolution of Figures 1 and 2 is very low and somewhat difficult to read. It is important that our Editors and Peer Reviewers are able to read all parts of a submission. Please replace these figures with higher resolution copies.

Additional Editor Comments (if provided):

Reviewers' comments:

Reviewer's Responses to Questions

**Comments to the Author**

1. Does this manuscript meet PLOS Global Public Health’s publication criteria ? Is the manuscript technically sound, and do the data support the conclusions? The manuscript must describe methodologically and ethically rigorous research with conclusions that are appropriately drawn based on the data presented.

Reviewer #1: Yes

Reviewer #2: Yes

2. Has the statistical analysis been performed appropriately and rigorously?

Reviewer #1: I don't know

Reviewer #2: No

3. Have the authors made all data underlying the findings in their manuscript fully available (please refer to the Data Availability Statement at the start of the manuscript PDF file)?

Reviewer #1: Yes

Reviewer #2: Yes

4. Is the manuscript presented in an intelligible fashion and written in standard English?

Reviewer #1: Yes

Reviewer #2: Yes

5. Review Comments to the Author

Reviewer #1: Review report

Introduction

The authors provide an informative introduction to the study including key maternal health statistics at national and subnational level. The authors also highlight some elements related to referral care including issues arising from pre-referral care. However, a few elements seem inadequately addressed in the introduction, which the authors should consider clarifying. 1) referral care continuum? This is a key concept in their title but not very well explained in the introduction? What is the scope? Is it a hierarchy? Etc. 2) complex obstetric patients? A sentence or two clarifying this, in light of referral and other issues would be helpful.

Other specific points for consideration,

L45, suggest removing ‘antenatal’ because it should be part of “maternal health care”?

L47, authors should specify ‘skilled attendants’ where? Is it in Malawi or SSA?

Authors should improve coherency between the first and second paragraph. There is a disconnect. Perhaps rephrase L57 to be specific to the facility delivery coverage?

L60, It could be helpful for authors to specify where the pre-referral challenges were reported from? Which type and level of facilities?

L66, ‘exploration of barriers and bottlenecks in other areas….? Which areas? Geographical? Clinical? Referral continuum??? It would be helpful for authors to clarify.

L90, perhaps the third word should be “understand” instead of understanding?

Methods

L105, the section “setting and participants” seems to only include setting?

L125-126, the definition for “referral interval” seems inconsistent. “…..to refer to the charts in this sub-analysis involving maternal charts which……”

L130, could authors specify period referred to as “pre-COVID” charts?

Under researcher characteristics – authors state that they used the word “woman/women”… but in the methods they seem to use “patients”? is this different?

Authors should clarify on how the referral interval was computed, its not very clear as is

Results

The authors provide a range of results with good interpretation. A few things to point out for authors to consider. Overall it is not very easy to follow all the results presented, and there is inconsistent use of terminologies that are also not defined by the authors. Also, there are inconsistencies in variable nomenclature across tables in the results section, that is, one seemingly similar variable labelled differently in different tables. Perhaps providing a clearer scope of the work might help refine the presentation and consistency of the results. As previously highlighted, authors do not clarify how they operationalised “referral continuum” hence the range and mix of results that currently makes it hard to appreciate. What is the focus? What are the key elements of interest? Even the qualitative results could use some refining. It would be helpful for the results to sort of show or apply to the continuum instead of the way they are now mixed. Let us appreciate the issues with pre-referral care, then go on to understand (the consequences of this) reflected by quality of care at the receiving facility?

There seems to be a number of goal-posts that are not synchronised – ‘characteristics of pre-referral care driving poor outcomes’; ‘characterise obstetric care and outcomes across referral continuum and explore facilitators and barriers to the referral process’; ‘quality of pre-referral care and health outcomes’; ‘facilitators and barriers to timely quality obstetric referrals’……Authors should harmonise these, provide a clear scope then arrange results within that scope.

L212, the term ‘intrapartum period’ has not previously been defined

L215, the Fisher’s exact tests revealed no significant differences in what?

Authors should consider revising title of Table 1

Table 1, the variable ‘facility’, should it be ‘referring facility’ with another word to speak to the data there i.e., distance?

A number of terms or variables in Table 1 have not been defined

What is other ward in table 1?

L220, what is meant by “referral characteristic”?

Table 2, authors should clarify which statistic the p-value refers to

L234, it would be helpful for authors to clarify reference for establishing that r=0.2 is ‘moderate correlation’?

The section on the quality of data from the charts, L287-295 should come earlier, perhaps after participant characteristics.

L311, I would suggest authors to expand the current quote or find an alternative quote. The present one does not seem to fit well with the point put forward on long wait times and poor quality of care

L314, presuming that this text is in reference to QECH, I suggest the authors use referring facilities (for the lower-level facilities) and receiving facility (for what at the moment I think is termed as ‘referral hospital?’) to avoid confusion

L339, the sentence “participants perceived an insufficient supply of ambulances in high demand” is not clear, authors should consider revising it

L361, the authors should clarify in brackets, the “it” referred to in the quote

Discussion

Authors do provide ample explanation for the their results. Although consistency especially of the first two or three paragraphs could be improved. Also, generally the implications of the findings are inadequately addressed. Beyond Malawi, what should other SSA countries take from the study? But also interesting the “framework” that has been used for the discussion, which is not in methods or results. Authors should look into such inconsistencies… Why is referral interval not discussed? It seemed an important element of the study?

L395, confirmed???

L405-407, wondering why authors haven’t included work from other sub-Saharan African countries that have demonstrated the effect of distance and travel time with poor maternal and perinatal health outcomes? Perhaps they should also specify where the work referred to is underway?

L508, “Impact of social support” – this sections seems to have a mix of social and health system elements which I suggest be separated. The part of patients not being given clear reasons etc from L514, should be put in the section on communication, it is a measure of quality of care provided by the providers/system?. then, based on the narrative of “social support”, it seems more of social influence hence authors should specify that given that social support is broader than social influence.

L524, which component of care? ….this further speaks to the inconsistent use of terminologies or clear operationalisation of key constructs. Please harmonise

L526, the first recommendation – considering populations needs were not assessed in the present study, so based on challenges identified, could the authors be clear on what “especially increasing nurse-midwives” would help solve.

L539, the link between supplies and documentation is not clear, and was not reflected in their results – the results mostly linked it to patient load?

The first and last bullets under the suggestions of L533, can’t they be merged? Its all training, so specify what training for who and by who. And the text that follows it from L543? Can’t these be integrated?

The implication(s) of the first limitation does not come out clearly

L584, the suggestion is not very clear

Reviewer #2: Comments:

1) Introduction: I would make first paragraph two paragraphs. First paragraph dedicating problem statement: despite universal access to SBA, maternal and neonatal mortality is still high. Second paragraph would be dedicated to possible challenges for that. Third paragraph could be national efforts to address those challenges. Then the fourth and fifth paragraphs could be rationale and significance of the study.

2) Methods

2.1. What is the theoretical foundation of the study: the Three Delays Model or the 5S Model? Clarifying this would be easier if the researchers developed a conceptual framework illustrating the relationship between outcome and predictor variables. Clearly define the outcome variable(s) of interest for the analytical part of the study (e.g., maternal condition on arrival, maternal death, and/or referral interval) and the predictor variables (e.g., facility-level and individual-level predictors).

2.2. It’s great that mixed methods research is being used. Please indicate where both qualitative and quantitative methods were integrated—specifically in the Results or Discussion sections.

2.3. Have you assessed whether the sample size is adequate for subgroup analysis?

2.4. Please describe how the researchers ensured the trustworthiness of the qualitative components of the study, either in the Methods section or the Discussion section.

2.5. Analysis: It is unclear why multivariable regression accounting for clustering was not performed. Please conduct a multilevel regression analysis and report regression coefficients and 95% confidence intervals to identify predictors.

3. Results

3.1. To improve the distribution of values in each cell, I would recategorize several variables. For instance, in Table 2, consider recategorizing the following variables: pre-referral diagnosis, diagnosis upon admission, hospital procedures, and medications given.

3.2. In Table 2, the variables "distance to hospital" and "average hours to see provider" are likely to be correlated with each other. Similarly, in Table 5, the variables "facility" and "distance" may also be correlated. Please check for multicollinearity among these variables.

3.3. In Table 5, recategorize the "model of delivery" variable. Combine "forceps" and "vacuum" into a single category labeled "assisted delivery," and categorize "breech delivery" under "spontaneous vaginal delivery" or "assisted delivery" as appropriate.

3.4. The qualitative findings did not clearly identify the barriers and facilitators of pre-referral practice as outlined in the study's objectives. Additionally, I would expect to see an analysis of provider behavior as either a barrier or facilitator of referral practices.

4. Discussion: A bit of reorganization is needed. Summarize key finings in the first paragraph then interpret and discuss its implications in subsequent paragraphs and describe “triangulation of the findings” in the methods section.

6. PLOS authors have the option to publish the peer review history of their article (what does this mean? ). If published, this will include your full peer review and any attached files.

**Do you want your identity to be public for this peer review?** For information about this choice, including consent withdrawal, please see our Privacy Policy .

Reviewer #1: No

Reviewer #2: **Yes: ** Gizachew Tiruneh

---

## [Decision Letter · Decision Letter 1]

PGPH-D-24-00015R1

Characterizing the referral care continuum among complex obstetric patients in the Blantyre District of Malawi: A mixed methods study

Dear Dr. Mitchell,

Thank you for submitting your manuscript to PLOS Global Public Health. After careful consideration, we feel that it has merit but does not fully meet PLOS Global Public Health’s publication criteria as it currently stands. Therefore, we invite you to submit a revised version of the manuscript that addresses the points raised during the review process.

We look forward to receiving your revised manuscript.

Kind regards,

Annesha Sil, Ph.D.

Staff Editor

PLOS 

Additional Editor Comments (if provided): 

Reviewers' comments:

Reviewer's Responses to Questions

**Comments to the Author**

1. If the authors have adequately addressed your comments raised in a previous round of review and you feel that this manuscript is now acceptable for publication, you may indicate that here to bypass the “Comments to the Author” section, enter your conflict of interest statement in the “Confidential to Editor” section, and submit your "Accept" recommendation.

Reviewer #3: (No Response)

2. Does this manuscript meet PLOS Global Public Health’s publication criteria ? Is the manuscript technically sound, and do the data support the conclusions? The manuscript must describe methodologically and ethically rigorous research with conclusions that are appropriately drawn based on the data presented.

Reviewer #3: Partly

3. Has the statistical analysis been performed appropriately and rigorously?

Reviewer #3: Yes

4. Have the authors made all data underlying the findings in their manuscript fully available (please refer to the Data Availability Statement at the start of the manuscript PDF file)?

Reviewer #3: No

5. Is the manuscript presented in an intelligible fashion and written in standard English?

Reviewer #3: Yes

6. Review Comments to the Author

Reviewer #3: Thank you for the opportunity to review this manuscript. Overall, it offers valuable insights, though there are a few areas that could be strengthened. While some of these points were highlighted by previous reviewers, there are additional aspects that could benefit from further refinement.

Some suggestions and comments below:

Introduction:

- Line 49 pg4: the sentence ends abruptly, what solutions are referred to here?

- Line 57-60 pg 4: missing punctuation

- Lines 96-101 pg. 6: please revise the text to more clearly outline the study's primary aim and specific objectives. Additionally, we encourage you to emphasise how the study contributes to the broader scientific knowledge and clarify its potential relevance and value beyond the specific context of this location, as suggested also by previous reviewers.

Methods section: there are several aspects of the methods section that would benefit from further clarification. This is important for ensuring transparency and adhering to international standards in reporting practice (please refer to relevant EQUATOR network guidelines). Here are some examples:

- What methodological orientation underpins the qualitative aspect of the study?

- Could you please clarify the rationale for using both interviews and focus groups in your study, and how participants were assigned to each method? It would be helpful to understand the specific purpose of each approach and the criteria guiding participant allocation.

- Line 185-186: for transparency, please provide more detail on the specific variables included in the quantitative analysis, for example, which maternal care practices and outcomes were considered? And what was the rationale behind the selection of variables to include in the model?

Discussion:

- The discussion section is currently quite lengthy (10 pages). Please streamline it to focus more directly on the key findings and their implications.

Please consider moderating the recommendations to ensure they are more closely aligned with the study's findings. At times, they appear to reflect personal opinions rather than being directly supported by the data.

7. PLOS authors have the option to publish the peer review history of their article (what does this mean? ). If published, this will include your full peer review and any attached files.

**Do you want your identity to be public for this peer review?** For information about this choice, including consent withdrawal, please see our Privacy Policy .

Reviewer #3: No

---

## [Decision Letter · Decision Letter 2]

Characterizing the referral care continuum among complex obstetric patients in the Blantyre District of Malawi: A mixed methods study

PGPH-D-24-00015R2

Dear Ms. Mitchell,

We are pleased to inform you that your manuscript 'Characterizing the referral care continuum among complex obstetric patients in the Blantyre District of Malawi: A mixed methods study' has been provisionally accepted for publication in PLOS Global Public Health.

Best regards,

Julia Robinson

Executive Editor

Reviewer Comments (if any, and for reference):

Reviewer's Responses to Questions

**Comments to the Author**

1. If the authors have adequately addressed your comments raised in a previous round of review and you feel that this manuscript is now acceptable for publication, you may indicate that here to bypass the “Comments to the Author” section, enter your conflict of interest statement in the “Confidential to Editor” section, and submit your "Accept" recommendation.

Reviewer #3: All comments have been addressed

2. Does this manuscript meet PLOS Global Public Health’s publication criteria ? Is the manuscript technically sound, and do the data support the conclusions? The manuscript must describe methodologically and ethically rigorous research with conclusions that are appropriately drawn based on the data presented.

Reviewer #3: Yes

3. Has the statistical analysis been performed appropriately and rigorously?

Reviewer #3: Yes

4. Have the authors made all data underlying the findings in their manuscript fully available (please refer to the Data Availability Statement at the start of the manuscript PDF file)?

Reviewer #3: No

5. Is the manuscript presented in an intelligible fashion and written in standard English?

Reviewer #3: Yes

6. Review Comments to the Author

Reviewer #3: Thank you for the revisions, the manuscript is now clearer.

7. PLOS authors have the option to publish the peer review history of their article (what does this mean? ). If published, this will include your full peer review and any attached files.

**Do you want your identity to be public for this peer review?** For information about this choice, including consent withdrawal, please see our Privacy Policy .

Reviewer #3: No
